# Crop-Derived Biochar for Removal of Alachlor from Water

**DOI:** 10.3390/ma17235788

**Published:** 2024-11-26

**Authors:** Iwona Zawierucha, Jakub Lagiewka, Aleksandra Gajda, Jolanta Kwiatkowska-Malina, Damian Kulawik, Wojciech Ciesielski, Sandra Zarska, Tomasz Girek, Joanna Konczyk, Grzegorz Malina

**Affiliations:** 1Institute of Chemistry, Jan Dlugosz University in Czestochowa, Armii Krajowej 13/15, 42-200 Czestochowa, Poland; jakub.lagiewka@doktorant.ujd.edu.pl (J.L.); d.kulawik@ujd.edu.pl (D.K.); wc@ujd.edu.pl (W.C.); s.zarska@ujd.edu.pl (S.Z.); t.girek@ujd.edu.pl (T.G.); j.konczyk@ujd.edu.pl (J.K.); 2Strata Mechanics Research Institute, Polish Academy of Sciences, 30-059 Krakow, Poland; aleksandra.gajda@imgpan.pl; 3Faculty of Geodesy and Cartography, Warsaw University of Technology, Pl. Politechniki 1, 00-661 Warsaw, Poland; jolanta.kwiatkowska@pw.edu.pl; 4Department of Hydrogeology and Engineering Geology, AGH University of Krakow, Mickiewicza 30, 30-059 Krakow, Poland; gmalina@agh.edu.pl

**Keywords:** biochar, wheat grains, alachlor, water treatment

## Abstract

The presence of various pesticides in natural streams and wastewater is a significant environmental issue due to their high toxicity, which causes harmful consequences even at low quantities. One cost-effective method to remove these pollutants from water could be through adsorption using an inexpensive, easily obtained adsorbent—biochar. The presented research demonstrates the efficacy of applying biochar obtained from wheat grains to eliminate alachlor from water. The sorption properties of the biochar and the likely removal mechanisms are defined. The study found that the biochar removed 76–94% of alachlor, depending on the initial concentration of the pesticide in water. The maximum removal of alachlor (94%) using biochar occurred at an initial pesticide content of 1 mg/L. Both the pseudo-second-order kinetic (R^2^ = 0.999) and the Langmuir isotherm models (R^2^ = 0.996) effectively characterized the elimination of alachlor by wheat grain biochar. The biochar’s maximum adsorption capacity for alachlor was 1.94 mg/g. The analyzed biochar, with its micropores and various surface functional groups, was able to effectively adsorb alachlor and trap it within its structure.

## 1. Introduction

Pesticides, including herbicides, insecticides, nematicides, and fungicides, are widely used to increase crop production. However, these chemicals are toxic and harmful to both human health and the environment [1]. The occurrence of pesticides in surface and groundwater across the world has been reported in many publications during recent decades and reviewed by many authors [2,3,4,5,6,7,8,9], demonstrating an increasing concern. The worldwide rise of pesticide application (including herbicides) poses the risk of increased pollution to aqueous systems [10,11,12]. The behavior of pesticides in the environment is determined by relevant processes occurring during their transport between defined sources and receptors via indicated pathways, e.g., on the surface (run-off), between the topsoil and atmosphere, and between surface waters and shallow groundwater systems. The impact of urban areas (e.g., wastewater treatment plants) on pesticides’ appearance in the aquatic ecosystems should also be accounted for [13].

Alachlor (CAS No.: 15972-60-8) is a chloroacetanilide herbicide (molecular formula: C_14_H_20_ClNO_2_) that is usually applied to combat broadleaf weeds and annual grasses before they emerge in crops like maize, sorghum, and soybeans [14] as well as in cotton, brassicas, oilseed rape, peanuts, radish, and sugar cane [15,16]. Alachlor is the common name for 2-chloro-N-(2,6-diethylphenyl)-N-methoxymethylacetamide. It is a white odorless crystalline solid at 23 °C, with a molecular weight of 269.8, water solubility of 242 mg/L at 25 °C, vapor pressure of 2.9 × 10^−3^ Pa at 25 °C, a log octanol–water partition coefficient of 2.6–3.1 [17], and a half life of 7–38 days in soil [18]. In addition to its slight solubility in heptane, alachlor is also soluble in ether, acetone, benzene, chloroform, ethanol and ethyl acetate.

After its application to plants, alachlor disappears from soil primarily as result of volatilization, photodegradation, and biological degradation [19]. In specific circumstances, alachlor can penetrate beyond the root zone and further to shallow groundwater [16]. Some other environmental transport paths of alachlor include direct drainflow, surface runoff, and irrigation to surface waters, as well as discharge from wastewater treatment plants [13]. This herbicide irritates mucous membranes, but the accurate mechanism of possible teratogenic alterations is still under study. In mammals, alachlor forms conjugates with glucuronic, sulfate, and mercapturic acids. Sister chromatid exchanges have been observed in human lymphocytes in vivo, and in vitro as dose-dependent chromosomal aberrations [20].

Existing experimental research results do not clearly indicate the genotoxicity of alachlor. On the other hand, metabolites of alachlor have been found to be mutagenic, while some other data clearly denote its carcinogenic nature, leading to benign and malignant tumors of the nasal turbinate, malignant stomach tumors, and benign tumors of the thyroid gland. The observed oral dose considered lethal in humans is 0.5–5 g/kg body weight [21]. Based on the NFPA-704 M rating system, the hazard identification of alachlor is as follows: health 2, flammability 0, reactivity 0, low solubility in water [17]. According to the World Health Organisation (WHO), the greatest permissible amount of alachlor that may be found in drinking water is 0.02 mg/L [22].

Pesticide contamination of surface and groundwater remains a problem due to non-point sources, such as agricultural runoff [23]. In order to remove pesticides from water, a number of different techniques may be applied. In addition to adsorption, these techniques include photocatalytic decomposition, oxidation with hydrogen peroxide or ozone, advanced oxidation, application of membranes, electrochemical destruction, coagulation, flocculation, biological treatments, and hybrid technologies [1,24]. Compared with other methods, adsorption is a favored option for pesticide removal due to its simplicity, operational ease, versatility, and high efficiency. Moreover, adsorption does not produce any toxic byproducts [24].

Adsorption using inexpensive, readily available adsorbents could be the most cost-effective option for removing emerging contaminants from water. Biochar, an environmentally sustainable and cost-effective material typically derived from organic waste such as agricultural residues, wood byproducts, and municipal waste, has garnered significant interest due to its relevance in the context of a circular economy [25,26]. Due to its properties, such as high carbon content, specific surface area, and cation/anion exchange capacity, and its durable structure [27,28,29,30,31], biochar has been reported as a highly efficient material for removing various inorganic pollutants including heavy metals [32,33,34,35,36]. Biochar has also been approved as a likely sorbent to reduce organic pollutants in water and soil [37,38]. Jagadeesh and Sundaram (2023) provided a comprehensive review of papers on the removal of microplastics, nutrients, and organic pollutants (fertilizers, antibiotics, PAHs, and PCBs) [39]. The physico–chemical properties of biochar determine its capacity to remove pollutants from aqueous solutions. These properties depend on the feedstock, thermal conversion technique, and preparation conditions. The relatively low raw material cost, easy production procedure, and suitable physico–chemical properties make biochar suitable for treating polluted aqueous streams [36]. Various modification methods are available to produce biochar-based composite materials with better adsorption properties compared with untreated biochar. These include magnetic, acid, alkali, steam, and nano-metal oxide/hydroxide modifications. Such modifications can improve the specific surface area, porosity, and surface functional groups of biochar. The relevant mechanisms of biochar adsorption mainly include pore filling, electrostatic interactions, hydrogen bonding, hydrophobic interactions, and π–π junctions, and they may differ as a result of, e.g., feedstock, pyrolysis temperature, kinetic parameters, coexisting ions, and biochar particle size [40].

Crop residue-based biochars, mainly prepared from straw, have been studied as adsorbents for pesticide removal from water. The biochar materials derived from corn straw were successfully prepared and tested for their ability to remove carbendazim from the water environment, by Wang et al. (2022) [41]. It was noted that the biochar surface area, pore structure, and functional groups were all positively affected by the pyrolysis temperature, which in turn accelerated the adsorption of carbendazim. The highest adsorption capacity of carbendazim (108.1 mg/g) was obtained with corn straw biochar prepared at 700 °C and modified with FeCl_3_. Ćwieląg-Piasecka et al. (2023) conducted a study on the sorption of pesticides on pristine and deashed biochar [42]. They found that hydrophobic pesticides (metolachlor and carbamates) exhibited high adsorption rates of 88–98% on both biochars. The study by Mandal and Singh (2017) on the removal of atrazine and imidacloprid from polluted water with pristine and phosphoric acid-treated rice straw biochar [43] showed the materials’ meaningful potential to adsorb both pesticides. In turn, Okoya et al. (2020) found that rice husk biochar effectively removed chlorpyrifos from water, with a removal rate of approximately 94% [44].

After reaching adsorption saturation, carbonaceous materials become solid waste and may pose the environmental threat of contaminant leaching, resulting in secondary pollution [45]. Therefore, the regeneration of saturated sorbents is of great environmental concern and has significant economic value. The most widely used thermal regeneration process leads to high energy demands and costs of transportation, as well as pollutant gas emissions [46]. A number of studies have focused on chemical regeneration [47], researching new regeneration techniques for saturated carbonaceous materials, e.g., ultrasonic [48], microwave [49], biological [50], and supercritical water or carbon dioxide regeneration [51]. Vacuum and electrochemical regeneration methods have also been considered [52,53].

This study demonstrates the effectiveness of alachlor removal from water using biochar derived from wheat grains. It examined the sorption characteristics of the biochar, including kinetics and isotherms, and assessed any alterations in biochar properties following alachlor sorption to understand the potential removal mechanism. Biochar produced from wheat grains can function as an effective and affordable sorbent for removing alachlor from polluted water, while also offering a solution for the disposal of agricultural waste.

## 2. Materials and Methods

### 2.1. Chemicals

The potassium bromide (FT-IR grade, ≥99%), ethanol (96%) and alachlor of analytical standard were provided by Merck KGaA, Darmstadt, Germany. Deionized water was used to prepare the aqueous solutions.

### 2.2. Preparation of Biochar

Wheat grains originated from the Silesian Department of Grain in Czestochowa, Poland. After initial treatment with tap water to remove water-soluble impurities, the grains were washed with demineralized water and dried in an oven at 105 °C for 24 h. The prepared samples were put in a porcelain crucible and inserted in a proportional–integral–derivative (PID)-control muffle furnace at 650 °C (20 °C/min) for 60 min. under limited oxygen availability (argon atmosphere) with a gas flow of 50 mL/min. The obtained samples were cooled and placed in desiccators to avoid further moisture absorption. A sieve with a mesh diameter of one millimeter was then used to pass the material through after it had been ground. The biochar was kept at room temperature in a bottle made of glass until it was required for use.

### 2.3. Determination of Biochar Characteristics

The elemental analysis of the pristine biochar was conducted using a FlashSmart CHNS/O Elemental Analyser—Thermo Fisher Scientific apparatus (Thermo Fisher Scientific, Waltham, MA, USA). The biochar pH was measured using a Fisher Scientific Accumet AR50 pH-meter (Thermo Fisher Scientific), following the procedure described by Li et al. (2013) [54].

The FT-IR analysis was performed using a Thermo Nicolet Nexus ((Thermo Fisher Scientific, Waltham, MA, USA) with the KBr pellet method. The biochar was examined before and after the sorption of alachlor. The spectral analysis focused on shifts and intensity variations of absorbance bands associated with the material’s functional groups, to investigate the molecular mechanisms of the sorption process.

Morphological analysis of biochar samples (before—BC and after alachlor sorption—SBC) was conducted using a VEGA3 TESCAN scanning electron microscope (TESCAN, Brno, Czech Republic). The samples were prepared by affixing them to a metal stub using an adhesive tube. The examination of the uncoated sample was conducted under high-vacuum conditions. The photomicrographs of the biochar were captured utilizing secondary electrons and an accelerating voltage of 10 kV.

To establish the pore structure of BC and SBC biochar samples, low-pressure gas adsorption was analyzed on an ASAP 2020 (Micromeritics, Norcross, GA, USA). This method describes the structure of microporous materials well and is used to characterize micropores and small mesopores in the material structure. Nitrogen was used as the adsorbate [55,56]. The measurement consisted of registering the volume of gas adsorbed on the sample surface. The measurements were carried out in isothermal conditions at the temperature of liquid nitrogen (77 K) and at absolute pressure in the range of 0–100 kPa and relative pressure in the range of 0 < p/p^0^ < 0.996, which is the ratio of absolute pressure to critical pressure of nitrogen in the gas phase. Before measurement, the samples were degassed for 12 h in UHV at 363 K. The Langmuir, Brunauer–Emmet–Teller (BET) and NLDFT models were used to describe the structural biochar parameters. The Langmuir model is a single-layer adsorption model. This model assumes the existence of active sites on the surface of the material, each of which can adsorb only one adsorbate molecule, and the gas molecules do not interact with each other [57]. The BET model is based on the Langmuir model and describes multilayer adsorption [58]. The NLDFT model is based on density functional theory and enables the analysis of pore size distribution in the micro- and mesopore range [59].

### 2.4. Sorption Experiments

The sorption experiments (batch tests) were performed in a 25 mL sealed conical flask and an KS 4000 IC Control incubator shaker (IKA, Staufen, Germany). After shaking 50 mg of BC with 10 mL of alachlor solutions ranging from 1 to 10 mg/L for 24 h, the mixture was centrifuged at a speed of 12,000 rpm. The supernatant was then collected and filtered through a 0.45 µm membrane filter. The samples were analyzed using gas chromatography coupled with mass spectrometry to determine the pesticide content. To determine the effect of contact time on the sorption process efficiency, 50 mg of BC was agitated with 10 mL of alachlor solution (10 mg/L) for between 5 and 1440 min.

The following equations were used to describe the effectiveness of the biochar in terms of alachlor removal and alachlor uptake during the experiment:(1)$$\%\mathrm{removal}=\frac{C_{0}-C_{e}}{C_{0}}$$
(2)$$q_{e}=\frac{(C_{0}-C_{e})\times V}{m}$$
where *C*_0_ and *C_e_* (mg/L) are the initial and the final alachlor concentrations (mg/L), respectively, *q_e_* is the sorption capacity parameter describing the amount of alachlor uptake per 1 g of biochar (mg/g), *V* is the volume of solution (L), and *m* is the mass of sorbent (g).

The efficiency of the biochar for the sorptive removal of alachlor from water was also examined using a flow-through column test (using a Millipore column, Merck KGaA, Darmstadt, Germany; Tygon tubing, Saint-Gobain, La Défense Cedex, France; and a peristaltic pump from Manostat Carter, Cole-Parmer, Vernon Hills, IL, USA). A Millipore column (length 250 mm, internal diameter 10 mm) held 0.5 g of the biochar. The column was saturated by applying an upward flow of distilled water. After saturation was reached at a steady flow rate of 1.5 mL/min, the water was displaced by the upward flow of the alachlor solution (at a concentration of 10 mg/L). The effluents were collected using a spur at the top of the column and taken until the ratio of the initial concentration (*C*_0_) to the effluent concentration (*C_e_*) equaled 1.

The following equation was used to compute the total amount of adsorbed alachlor(*q_total_*; mg) in the column at a particular feed concentration and flow rate:*q_total_* = (*QA*)/1000(3)
where *Q* is the flow rate (mL/min) and *A* is the area under the breakthrough curve obtained by integrating the adsorbed concentration (*C_ad_*; mg/L) versus t (min).

The total quantity of alachlor sorbed (*q_total_*) per mass of biochar (m) after complete flow time defined the equilibrium uptake (*q_e_*) (or maximum adsorption capacity) in the column.
*q_e_* = *q_total_*/*m*(4)

To displace the unadsorbed alachlor, the biochar was rinsed with deionized water. To explore the possibility of biochar regeneration, ethanol was then used to elute the alachlor that had been adsorbed onto the material.

Analysis of the alachlor concentration in samples was carried out via gas chromatography coupled with mass spectrometry, in accordance with Hong and Lemley (1998) [60] with minor modifications, as follows. An Agilent GC model 8890 chromatography system equipped with an MSD single quad detector, model 5997B GC/MS, and fitted with an HP-5MS UI column (30 m × 250 μm × 0.25 μm) was used with helium as carrier gas at a flow rate of 2 mL min^−1^. The injector and MSD transfer line temperatures were maintained at 280 and 280 °C, respectively. The oven temperature program was as follows: 120 °C, hold 1 min, 15 °C/min up to 180 °C, hold 1 min; 20 °C/min up to 280 °C, post-run time 1 min.

MS parameters for quantitative analysis were as follows. The SIM acquisition type was used. The MS source and MS quad temperatures were kept at 320 and 150 °C, respectively. For the recognition of alachlor molecules, 4 m/z ions such as 45, 160.10, 180, and 268.9 were applied. The delay time was 5 min.

Agilent Mass Hunter Quantitative Analysis software version 10.2 was used for quantitative analysis of the alachlor content in samples.

### 2.5. Isotherm and Kinetic Model

The sorption of alachlor on the biochar was characterized using the Langmuir and Freundlich isotherms, which are shown in Equations (5) and (6), respectively:(5)$$\frac{C_{e}}{q_{e}}=\frac{1}{q_{max\;}K_{l}}+\frac{C_{e}}{q_{max}}$$
(6)$$\log\;q_{e}=log\;K_{f}+\frac{1}{n}\;\log\;C_{e}$$
where *q_max_* (mg/g) is the maximum amount of alachlor adsorbed on the biochar, *K_L_* is the Langmuir constant (L/mg), and *K_f_* (mg/g) and 1/*n* are Freundlich empirical constants.

Pseudo-first- and second-order kinetics were applied to characterize the sorption rate of alachlor on biochar. These models were expressed with the following equations, respectively:(7)$$\ln(q_{e}-q_{t})=\ln q_{e1}-k_{1}t$$
(8)$$\frac{t}{q_{t}}=\frac{1}{k_{2}q_{e2}^{2}}+\frac{t}{q_{e2}}$$
where *t* is the time of contact (min), *q_t_* is sorption capacity at given time *t* (mg/g), and *q_e_* is sorption capacity at equilibrium (mg/g), while *k*_1_ and *k*_2_ are the rate constants of the pseudo-first (1/min) and pseudo-second (g/(mg·min) kinetic models, respectively.

## 3. Results and Discussion

### 3.1. Physicochemical Characteristics of Biochar

The pH of wheat grain biochar was alkaline (9.4), and its elemental composition was as follows (in %): C—74.3, N—12.7, H—6.1, and O—6.9. The contents of C, H, and O fell within the range typical of non-woody biochars, whereas the N content somewhat high relative to the literature data [61]. The appearance of polysaccharides and proteins in feedstock may be a probable reason for the occurrence of nitrogen in the studied biochar [29].

### 3.2. Performance of Wheat Grain Biochar in Removing Alachlor from Water

The results of alachlor removal over time are shown in Figure 1. As the contact time increased, the efficacy of alachlor removal also increased, reaching equilibrium after 180 min. At that point, the removal efficiency was 75%; however, more than 44% of the alachlor was rapidly adsorbed within 5 min.

The efficiency of the biochar in removing the alachlor also depended on the initial concentration of the pesticide in the water (Figure 2). As the amount of alachlor in the tested solutions increased, its removal efficiency decreased. The highest removal efficiency (93.9%) was achieved for an initial alachlor concentration of 1 mg/L. Consequently, alachlor sorption increased with its increased concentration in water. At an initial alachlor concentration of 10 mg/L, its uptake by the biochar was 1.59 mg/g. This value was similar to those obtained by Ćwieląg-Piasecka et al. (2023) for metolachlor and carbamate sorption on pristine wheat straw biochar [42].

Figure 3 shows the breakthrough curve for alachlor adsorption by biochar. After 20 min, a breakthrough was observed. In turn, the column was loaded after 100 min. The estimated adsorption column capacity for alachlor was 1.56 mg per gram of biochar. This was comparable to the result obtained in a batch experiment with the same (10 mg/L) concentration of alachlor.

An adsorbent with strong adsorption capacity and favourable regeneration capabilities might significantly reduce the total cost of the water treatment process. The findings of the investigations into regeneration showed that ethanol is sufficiently effective for this operation with spent biochar and may be used as a regenerative factor. After passing 20 mL of regeneration solution through the column, 81% of the alachlor had been desorbed. Apart from ethanol, typical eluents for effective herbicide desorption include NaOH aqueous solutions and acetonitrile. However, high desorption efficiency does not guarantee successful regeneration. Regeneration entails repairing biochar’s adsorptive characteristics after usage in order to renew its capacity for repeated adsorption cycles [62].

### 3.3. Isotherm and Kinetic Models

The sorption isotherms are shown in Figure 4a,b, corresponding to the Langmuir and Freundlich models, respectively. The model parameters are listed in Table 1.

The Langmuir isotherm provides a better fitting plot, with an R^2^ value higher than that of the Freundlich isotherm (0.996 > 0.962). This suggests that uniform alachlor binding to the biochar was part of the removal process. A Freundlich constant (n) value greater than 1 indicates a favorable adsorption process.

Pseudo-first- and second-order kinetics are presented in Figure 5a and Figure 5b, respectively, and model parameters are listed in Table 2.

The rate of alachlor sorption on biochar is best described by the pseudo-second-order model with the R^2^ value of 0.999. The *q_e_*_2_ value, 1.592 mg/g, corresponds to the calculated adsorption for 10 mg/L of alachlor during 24 h of agitation. The results fit the second-order model and suggest that alachlor removal by wheat grain biochar involves interactions of the sorbent functional groups with the pesticide molecules [25,63].

Several different biochar-based adsorbents for removing alachlor from aqueous solutions have been described in the literature. Table 3 summarizes the reported adsorption capacities of these biochars for alachlor, including the findings from our study. As seen in this comparison, only the modified biochar has a considerably higher adsorption capacity for alachlor compared with the other materials.

### 3.4. The Alteration of Biochar Properties as a Result of Alachlor Sorption

Based on the sorption values as a function of pressure, nitrogen adsorption isotherms were plotted for the BC and SBC samples (Figure 6). Nitrogen adsorption isotherms were characterized as type I according to the IUPAC classification [67], characterizing microporous materials. For both biochar samples, very similar isotherms were obtained, consistent with the shape of the Langmuir isotherm (Figure 1). The isotherm for the SBC sample was lower than the isotherm for the BC sample, indicating the incorporation of alachlor into the structure of biochar and the filling of some of the active centers on its surface.

Structural parameters were determined according to the BET, Langmuir, and NLDFT models, and their values are given in Table 4. The value of the Langmuir total sorption capacity a_mL_ decreased from 34.27 cm^3^/g to 25.63 cm^3^/g as a result of alachlor sorption. The total pore volume obtained from the NLDFT model also decreased and for BC and SBC samples, it was V_NLDFT_ = 0.06 cm^3^/g and V_NLDFT_ = 0.04 cm^3^/g, respectively. The surface area values of the BC sample (Table 4) were within the low range obtained for biochars derived from wood and produced at similar temperatures [61]. Moreover, the biochar had a total pore volume value of less than 0.1 cm^3^/g—typical when assessed through N_2_-sorption analysis [68]. The BET specific surface area and total pore volume measurements of the examined “raw” material were comparable to those of wheat straw biochar [42].

The pore size distributions for BC and SBC samples according to the NLDFT model are presented in Figure 7.

In the SBC sample, a decrease in the volume of pores with diameters below 0.87 nm was observed compared with the BC sample. This indicated adsorption of alachlor into the biochar structure and thus, a decrease in the availability of pores. These findings were supported by SEM images. The BC structure contained numerous channels and macropores (Figure 8a), offering abundant binding sites for alachlor adsorption. Following adsorption, these active sites became filled with the alachlor molecules (Figure 8b). The structure of the biochar indicates that the analyzed adsorbent can be used for effective sorption and trapping of alachor in its structure. The main factor recognized for the elimination of alachlor using the examined biochar, among various mechanisms, was pore filling [26,69].

The FT-IR spectra of BC and SBC samples are illustrated in Figure 9. The first wide and intensive band observed around 3400 cm^−1^ corresponds to O-H stretching, indicating the presence of OH groups. This is consistent with the results of the elemental analysis of 6.9% [70]. In addition, the application of wheat grain from feedstock containing polysaccharides and proteins can result in the presence of oxygen in the biochar structure. Polysaccharides are especially rich in oxygen, which changes form in the biochar during pyrolysis, e.g., to OH. The presence of aliphatic hydrocarbons is confirmed by the C-H stretching observed as a small band at 2920 cm^−1^ [70,71]. However, the shoulder at 1650 cm^−1^ can indicate also C=C; thus these bands may be a combination of C=O and C=C. The band at 1050 cm^−1^ describes bonds originating from different functional groups due to presence of C, N, and O atoms in the structure e.g., C-O, C-O-C, C-N [71]. The small band showing deformative stretching of O-H groups provides further evidence of hydroxyl groups in the biochar structure.

It is evident that the interaction between biochar and alachlor is related to hydrogen bonding. The significant decrease in intensity of bands at 3400 cm^−1^ strongly supports the interaction between hydroxyl groups from biochar and nitrogen/oxygen atoms from alachlor. Furthermore, the involvement of carboxylate groups in hydrogen bonding is evident from the decreased intensity and sharpened shape of the band at 1050 cm^−1^.

## 4. Conclusions

Biochar made from wheat grain was effective in removing alachlor from water. The most effective removal of alachlor (93.9%) using biochar was achieved at an initial pesticide concentration of 1 mg/L. Increasing the contact time of the sorbent with the pesticide improved the treatment efficiency, reaching an equilibrium state after 180 min. The maximum adsorption capacity of the biochar for alachlor was 1.94 mg/g. The biochar sorption capacity of the column was 1.56 mg/g, and after the first cycle, alachlor was desorbed with ethanol, achieving an efficiency of 81%.

The microporous nature of the biochar facilitated the incorporation of alachlor into the biochar structure and the filling of active sites. The removal of alachlor by wheat grain biochar also included adsorption via the interaction of functional groups on the biochar surface with the pesticide molecules.

Sorption of alachlor on biochar reduces the movement of pollutants and the potential risk of environmental contamination. In addition, the use of this sorbent in water and wastewater treatment could provide a sustainable solution for the recycling of agricultural waste, which is important in the context of the circular economy. However, features such as biochar reusability, selectivity, improving sorption characteristics by modification, optimalization of column work parameters, and the fate of eluents containing desorbed pesticide should be addressed, since these elements provide considerable challenges in relation to the practical usage of this material.

## Figures and Tables

**Figure 1 materials-17-05788-f001:**
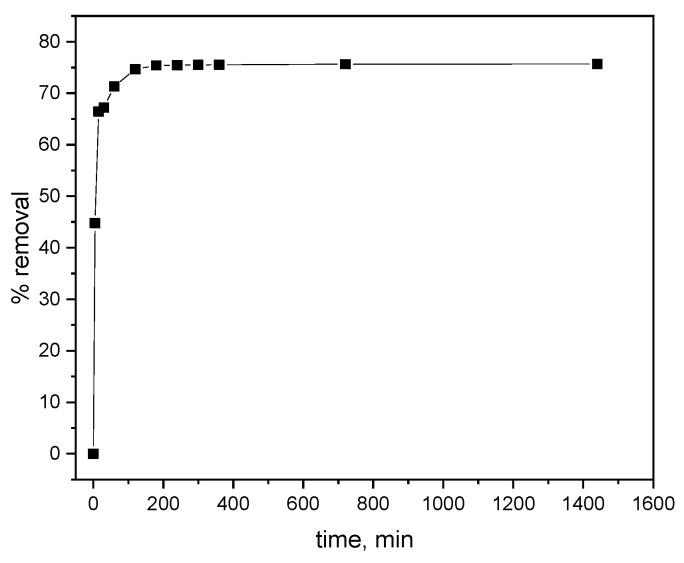
The effect of contact time on the removal of alachlor from water using biochar.

**Figure 2 materials-17-05788-f002:**
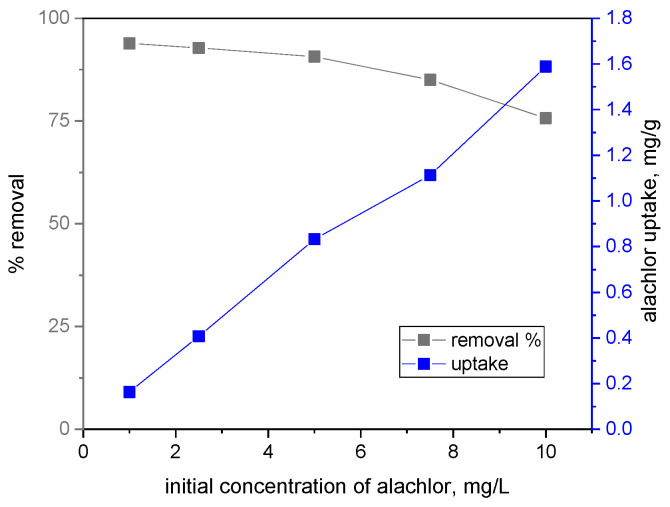
Efficiency of biochar for removing alachlor from water.

**Figure 3 materials-17-05788-f003:**
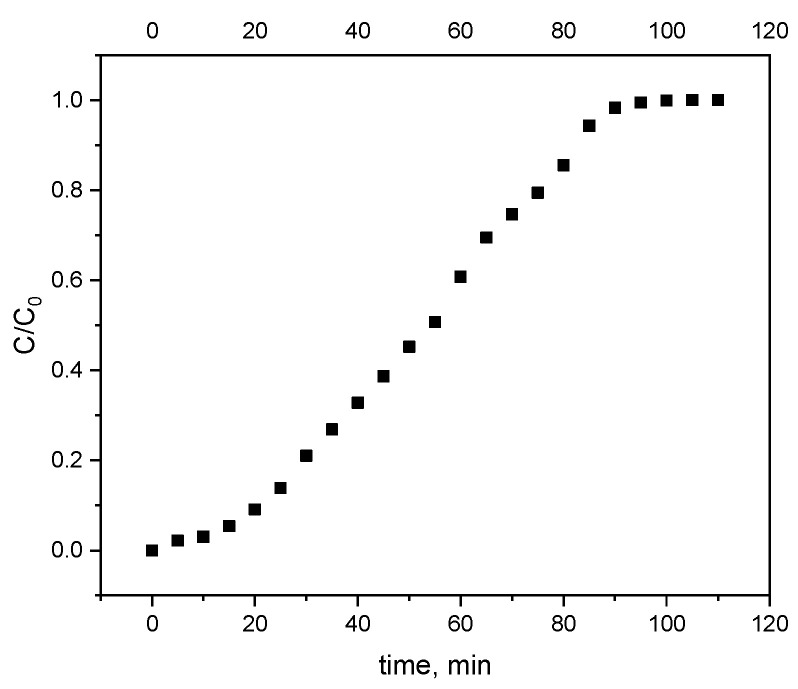
Breakthrough curve of alachlor adsorption by biochar.

**Figure 4 materials-17-05788-f004:**
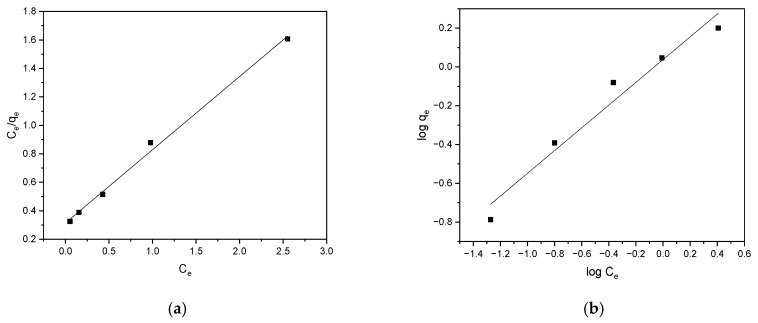
Langmuir (**a**) and Freundlich (**b**) isotherms for removal of alachlor by biochar.

**Figure 5 materials-17-05788-f005:**
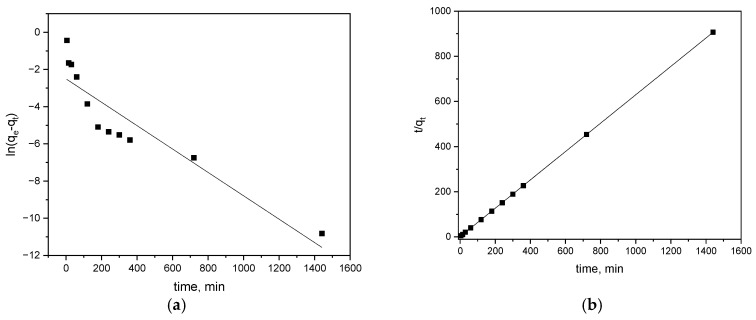
The pseudo-first- (**a**) and pseudo-second-order (**b**) kinetic models for removal of alachlor by biochar.

**Figure 6 materials-17-05788-f006:**
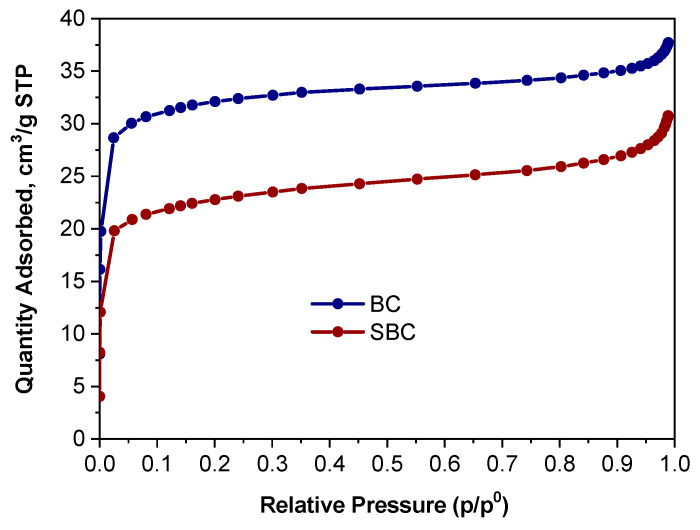
Nitrogen adsorption isotherms on BC and SBC samples.

**Figure 7 materials-17-05788-f007:**
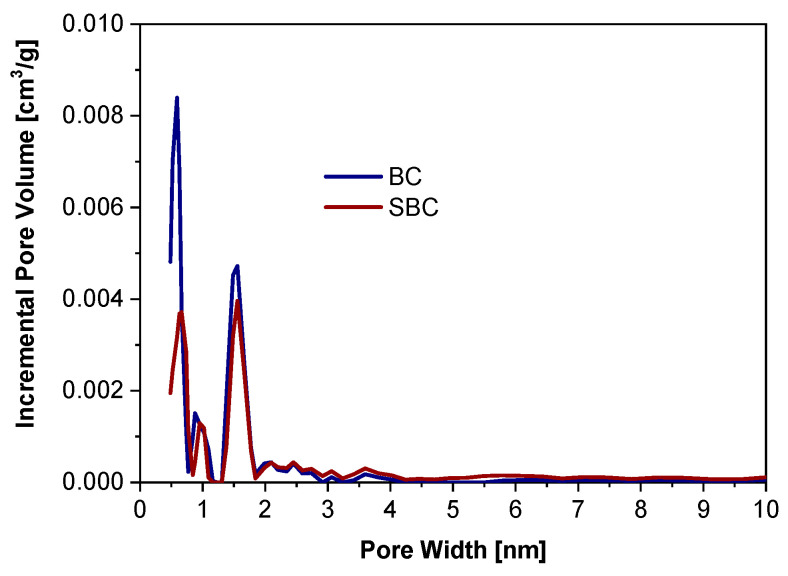
Pore size distribution for BC and SBC samples according to the NLDF model.

**Figure 8 materials-17-05788-f008:**
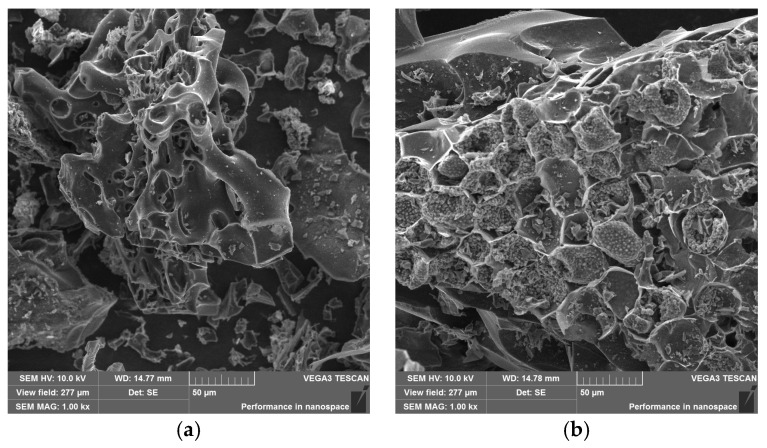
Structure of BC (**a**) and SBC (**b**) samples.

**Figure 9 materials-17-05788-f009:**
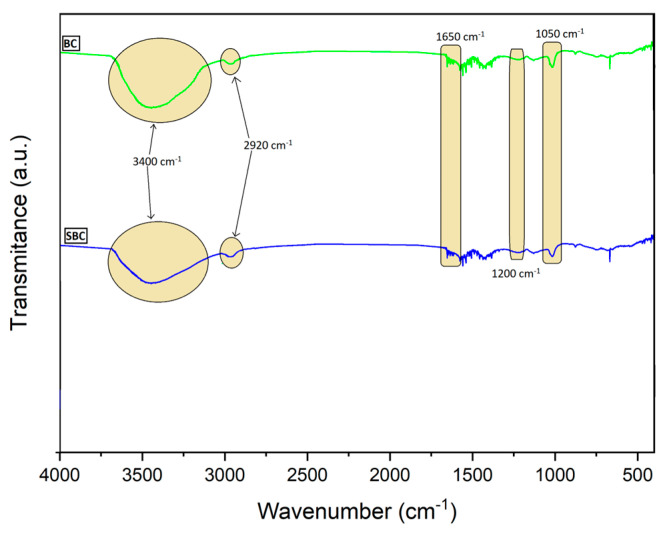
The FT-IR spectra of BC and SBC samples.

**Table 1 materials-17-05788-t001:** The isotherm parameters for sorption of alachlor on biochar.

Isotherm Models	Parameters	Values
Langmuir	*q_max_*	1.94
	*K* _l_	1.64
	R^2^	0.996
Freundlich	*n*	1.71
	*K_f_*	1.09
	R^2^	0.962

**Table 2 materials-17-05788-t002:** Parameters of the kinetic models for sorption of alachlor on biochar.

Kinetic Model	Parameters	Values
Pseudo-second order	*q_e_* _2_	1.592
	*k* _2_	0.271
	R^2^	0.999
Pseudo-first order	*q_e_* _1_	0.082
	*k* _1_	0.006
	R^2^	0.841

**Table 3 materials-17-05788-t003:** Comparison of adsorption capacity of different biochars for alachlor.

Biochar Feedstock	Adsorption Capacity (mg/g)	Reference
sugar beet shreds	0.10	[64]
Miscanthus giganteus	0.74	[64]
ground coffee residue (without activation)	3.24	[65]
ground coffee residue (with NaOH activation)	33.09	
leonardite	3.80	[66]
wheat grain	1.59	[This study]

**Table 4 materials-17-05788-t004:** Structural parameters of BC and SBC samples.

Low Pressure Nitrogen Adsorption, 77 K	Symbol	BC	SBC
Langmuir total sorption capacity, cm^3^/g STP	a_mL_	34.27	25.63
Langmuir coefficient, 1/kPa	K	0.85	0.48
Langmuir specific surface area, m^2^/g	SSA_L_	149.15	111.55
BET total sorption capacity, cm^3^/g STP	a_mBET_	22.61	16.28
BET specific surface area, m^2^/g	SSA_BET_	98.42	70.88
NLDFT total pore volume, cm^3^/g	V_NLDFT_	0.06	0.04

## Data Availability

The original contributions presented in the study are included in the article, further inquiries can be directed to the corresponding author.

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
