# Peer review of "Crop-Derived Biochar for Removal of Alachlor from Water"

_materials, 2024, doi:10.3390/ma17235788_

Round 1

Reviewer 1 Report

Comments and Suggestions for Authors

The authors describe the fabrication of biochar for water treatment. This biochar has been tested for the removal of pesticide alachlor from water. The manuscript is well-structured and clearly written. The data presented is supported by a significant number of references, most of which are up to date. Although the current research lacks novelty, it could provide valuable information to researchers in the field of water treatment.  However, there are some issues that need to be addressed in the manuscript.

Line 252: Based on the results obtained from fitting experimental data using the pseudo-second-order kinetic model, the authors claim that the main mechanism of adsorption is chemisorption. However, this claim is highly unlikely and is not supported by any other experimental data.

According to the BET investigation and pore size distribution data (section 3.4), it is concluded that the biochar is a microporous material, as all its pores are smaller than 2 nm. By comparing the data before and after adsorption, it can be concluded that the primary adsorption occurs in the micropores, as there is a significant decrease in pore volume. This suggests that adsorption is not mainly occurring on the surface of the biochar, contrary to the claims made by the authors.

Lines 293-308: FTIR analysis has been utilized to identify the presence of surface groups and potential interactions between biochar and the adsorbate. Comparing the peak intensities in spectra obtained using the KBr pellet method raises questions. The band appearing around 3400 cm-1 is attributed to -OH group stretching vibrations, but it may not solely originate from surface hydroxyl groups. It may also stem from moisture, particularly if the KBr method was used. The change in intensity of this band does not strongly suggest interactions between the biochar hydroxyl groups and alachlor. Further, a shoulder that appears at around 1650 cm-1, may be attributed to the C=C stretching vibration. Additionally, the band at 1050 cm-1 could originate from various oxygen-containing groups with a C-C-O structure. Since the pH measurement indicated that the biochar contains predominantly alkaline groups, the presence of a significant number of COOH surface groups is unlikely.

Figure 9 should illustrate the potential interactions between biochar surface groups and the alachlor molecule. In the micropores of carbonaceous materials, Van der Waals interactions significantly influence the bonding of the adsorbate. Specific interactions, such as hydrogen bonding, occur but are not solely responsible for the adsorption mechanism. This figure oversimplifies the investigated adsorption process and is somewhat inaccurate. Therefore, it would be advisable to remove it from the manuscript.

Reviewer 2 Report

Comments and Suggestions for Authors

Please see the attachment file (comments for materials-3285060).

Reviewer 3 Report

Comments and Suggestions for Authors

-In abstract: add more insights of the present study (qe value, max. removal, etc.)

-The plagiarism percentage is very high (37%), kindly check immediately and do the needful

-Two different kindly of referencing style is there??

-Continuously three paragraphs starting with the term "Alachlor", kindly merge them.

-mention the minimal permissible limit of Alachlor in the Introduction

-Mention other removal techniques as well, apart from adsorption and mention their limitations, before introducing the adsorption

-Mention the other chemicals as well in section 2.1

-Mention the reference for Alachlor quantification from HPLC

-In Fig 2, caption mentioned inside the figure should be removal% and not removal

-Compare with at least 2-3 more models as well, before the final conclusion

-redraw the FTIR spectra again with the baseline correction

-There should be substantial enhancement in the technical writing of the manuscript

-Manuscript totally lacks the discussion portion, add respection citations and discussion, elsewhere required

There is no section of either reuse or regeneration of the biochar. Nowadays, everyone has to do the complete study in regards with the column study and regeneration of the spent adsorbent

-Add the limitations and future scope of the present study

-The conclusion is very generalized

-Add a specific Table that include the biochar for Alachlor removal including the present study as well.

Comments on the Quality of English Language

The manuscript totally lacks the technical writing and discussion portion.

There is 37% plagiarism.  

Round 2

Reviewer 1 Report

Comments and Suggestions for Authors

The authors made an effort to enhance the manuscript. All issues from the original submission requiring revisions have been addressed.